# Predicting the Oxidative Degradation of Raw Beef Meat during Cold Storage Using Numerical Simulations and Sensors—Prospects for Meat and Fish Foods

**DOI:** 10.3390/foods11081139

**Published:** 2022-04-14

**Authors:** Alain Kondjoyan, Jason Sicard, Paolo Cucci, Fabrice Audonnet, Hiba Elhayel, André Lebert, Valérie Scislowski

**Affiliations:** 1Qualité des Produits Animaux, Institut National de Recherche pour l’Agriculture, l’Alimentation et l’Environnement, 63122 St.-Genès-Champanelle, France; alain.kondjoyan@inrae.fr (A.K.); hiba.elhayel@etu.isima.fr (H.E.); 2Institut Pascal, Université Clermont Auvergne, CNRS, 4 Av. Blaise Pascal, Campus Universitaire des Cézeaux, 63178 Aubière, France; paolo.cucci@uca.fr (P.C.); fabrice.audonnet@uca.fr (F.A.); andre-lebert@wanadoo.fr (A.L.); 3Institut d’Informatique d’Auvergne, Université Clermont Auvergne, 1 Rue de la Chebarde, Campus Universitaire des Cézeaux, TSA 60026, CEDEX, 63178 Aubière, France; 4Association pour le Développement de l’Institut de la Viande, 10 rue Jacqueline Auriol, 63100 Clermont-Ferrand, France; valerie.scislowski@adiv.fr

**Keywords:** food waste, modeling, kinetics, diffusion, color, myoglobin

## Abstract

Preventing animal-source food waste is an important pathway to reducing malnutrition and improving food system sustainability. Uncontrolled color variation due to oxidation is a source of waste as it prompts food rejection by consumers. Evaluation of oxidation–reduction potential (ORP) can help to predict and prevent oxidation and undesirable color changes. A new sensor and two modeling approaches—a phenomenological model and a reaction–diffusion model—were successfully used to predict the oxidative browning of beef ribeye steaks stored under different temperature and oxygen concentration conditions. Both models predicted similar storage durations for acceptable color, although deviating for higher and lower redness levels, which are of no interest for meat acceptance. Simulations under higher oxygen concentrations lead to a few days of delay in the redness change, as observed in practice, under modified atmosphere packaging. In meat juice, variation in ORP measured by the sensor correlated with the redness variation. However, in meat, sensors promote oxidation in the adjacent area, which is unacceptable for industrial use. This paper discusses the potential, limits, and prospects of the mathematical models and sensors, developed for beef. A strategy is proposed to couple these approaches and include the effect of microorganisms.

## 1. Introduction

World population growth and rising living standards are projected to increase demand for animal-source foods [1]. Livestock supports food security by converting feed, most of which is not human-edible, into food. Furthermore, in developing countries, increasing access to animal-source foods is an important pathway to reducing malnutrition, as animal foods are a valuable source of highly digestible and amino-acid-balanced proteins and micronutrients that can help counter nutritional deficiencies [2]. However, meat production requires more energy, water, and land than plant crops, farmed ruminants are a source of greenhouse gas emissions, and fish stocks are in decline. In this context, it is crucial to limit the loss and waste of meat and fish due to uncontrolled variations in color, which are a major source of food rejection by consumers.

Meat and fish food color is linked to multiple factors, among which are microbial spoilage, lipid oxidation, and protein oxidation (specifically, myoglobin oxidation) [3].

Microbial spoilage depends on technological treatment and storage conditions. Grinding remixes the surface area with the inside of the food and brings air into the mixture, which tends to promote microbial growth and oxidation within the product. In addition to the potential safety problems, the incorporation of spoilage microorganisms can accelerate the oxidation of the product. Grinding parameters themselves are able to change the color of the product, as already analyzed for pork meat [4]. Modified-atmosphere packaging (MAP) is considered as a way to limit bacterial growth and browning of the meat surface. In general, packaging affects color, spoilage, and oxidation and has more impact than breed and finishing feed on sheep meat color [5].

Lipid oxidation has been studied on many animal meats [5,6,7,8], and it is critical for foods rich in unsaturated fatty acids such as fish [8]. The lipid content of tuna varies according to species, but fish diet also strongly affects it. For this reason, antioxidant supplements are often added to the diet of farmed fish [9]. Similarly, the addition of antioxidants to the diet of cattle before slaughter is a strategy to decrease meat oxidation [10]. For beef, it has been shown that breed, sex, age, amount and nature of the lipids in muscle, and the presence of antioxidants linked to animal grazing affect meat oxidation and color [11,12]. Antioxidants can also be incorporated directly in processed meat products [13,14,15,16,17,18,19,20,21,22,23] or in the packaging [24,25,26], and the various antioxidant compounds represent a thriving area of research. Oxidation reactions, particularly lipid oxidation reactions, still develop at negative temperatures. Therefore, differences in freezing temperatures after the fishing of wild tuna may affect its flesh color [27]. Long periods of storage also affect consumer perception of meat redness [28].

Protein oxidation plays a role in meat product color [29,30,31], particularly the oxidation of myoglobin. The color of fresh beef, pork, or lamb meat is greatly affected by the presence of myoglobin, which in practice is mainly found in three forms: deoxymyoglobin (Mb), oxymyoglobin (MbO_2_), and metmyoglobin (MMb). Deoxymyoglobin lends meat a purplish color, oxymyoglobin lends meat a bright red color, and metmyoglobin lends meat a brown color. Consumers look to buy bright red color meat and reject brown meat [32,33]. This problem also concerns some fishes such as bluefin tuna, which is rich in myoglobin and whose color when sold raw, fresh or frozen, is an essential component of its quality [27].

Several reaction–diffusion models have been proposed over time to represent the phenomena implicated in myoglobin chemical state evolution. Saenz et al. [34] developed a simple linear model using only two equilibrium reactions to interpret the profiles of the different forms of myoglobin measured at different times during storage of a beef *Longissimus lumborum* (*LL*) muscle. Tofteskov et al. [35] developed a more comprehensive numerical model to predict meat preservation under modified-atmosphere packaging (MAP). Kondjoyan et al. [36] developed the most comprehensive model of myoglobin chemical state to date, which considered the kinetics of 22 impactful reactions.

Meat is a solid matrix that absorbs, scatters, reflects, and transmits light. Myoglobin and other pigments allow meat to absorb light in the Soret region of the visible spectrum (blue) and reflect light more intensively in the red region [37]. The resulting color depends on the characteristics of the incident light, the concentration of myoglobin at the meat surface, and the muscle structure. The color change in beef muscles is usually only measured via CIELAB redness coordinate *a**. The oxidation of biochemical media can be characterized by the variation of their oxidation–reduction potential (ORP), which is a thermodynamic variable, measured in volts (V). Commercial probes can be used to measure ORP in liquids but are not suitable for measuring ORP on a meat piece, because the imperfect contact of the electrodes with the meat surface leads to uncontrolled variation in the measure.

The development of artificial intelligence, a form of modeling, has driven and continues to drive substantial advances in many fields of science and technology. However, efforts to predict oxidation in foods do not stand to benefit much from this development due to the small amount of data available on the measure of oxidation kinetics under storage conditions. Therefore, this paper focuses on two other modeling approaches: phenomenological modeling and reaction–diffusion modeling. This paper shows how sensors and mathematical models can predict and, thus, prevent the oxidation of beef meat cuts responsible for their color change. The paper also discusses how the approach and results obtained on beef can be applied to other meats and to raw fish.

## 2. Materials and Methods

### 2.1. Meat Products

Beef ribeye roast boneless [38] samples from young cattle (Puigrenier slaughterhouse, Montluçon, France) were semi-dressed [39], vacuum-packed, and stored in the refrigerator at 4 ± 2 °C.

Beef ribeye steak boneless [38] samples were cut into 2 to 3 cm thick slices, weighing 350 to 400 g, deposited in sealable trays in PS/EVOH/PE (Form’plast, France), put under modified atmosphere, and then sealed with OPP/T504 film (Soussanna, France) thanks to a semi-automatic traysealer (T200, Multivac, France).

Meat juice was prepared from *Longissimus thoracis* (*LT*) muscles, the core muscle of the ribeye steak [40]. After reception, the muscles were cut into pieces 3 cm thick with a weight of about 80 g and packaged in polyamide/polyethylene (PA/PE) storage bags impermeable to gases (three welds, 200 × 300 mm, Sovapack, Cuiseaux, France). The bags were hermetically sealed either under half vacuum (500 mbar) or under vacuum with a double-chamber machine (Multivac, Lagny Sur Marne, France). The samples were then frozen at −20 ± 2 °C for 24 h and thawed at 4 ± 2 °C for 72 to 96 h depending on the size of the samples. The obtained juices after freezing and thawing of the samples were collected in a sterile manner under a microbiological safety station.

### 2.2. Sensor to Track the Time-Course of Oxidation

An in-house oxidation–reduction potential sensor, represented in Figure 1, was designed and built with two 1 mm diameter ‘working’ electrodes, one of which was a platinum rod (99.95% purity, Surepure Chemetals, Florham Park, NJ, USA) while the other was an oxidized iron rod [41].

The rods were connected to a digital data acquisition device using electric wires with a gap between the electrodes that was molded in resin. The electric signal was checked in various liquids, and the results were compared with the commercial probe (Hanna Instruments, Woonsocket, RI, USA). Commercial ORP sensors commonly use gels or electrolyte solutions, a platinum sensing pin, a ceramic junction (salt bridge), and a polyetherimide resin or glass body. Those designs make them suitable for measurement in liquids, but penetration into meat gives unreliable values and a risk of breakage. Using the probe designed by us, both electrodes could safely penetrate into the meat.

### 2.3. Color Measurements and Comparison with ORP

An industrial system (ADIV, Clermont-Ferrand, France) was used to track color changes in red meat. In a cold room (Dagard, Boussac, France) powered by a cold unit (Arcos, Gorrevod, France), cameras (Logitech C270 720p, Logitech, Paris, France) and strips of fluorescent tubes (T8, cool white color, 60 cm, 18 W, Sylvania, Saint-Etienne, France) were placed 30 cm above the meat trays. Strips were fixed on either side of the device with a distance of 45 cm between each strip, delivering approximately 1700 lux on the meat samples. Bitmap images of the samples (juice and meat) were taken every 15 min using software developed with LabVIEW 2018 and the Vision Development module (National Instruments Corporation, Austin, TX, USA). Using ImageJ software (National Institutes of Health, Bethesda, ML, USA), the values of lightness (*L**), red/green index (*a**), and blue/yellow index (*b**) were calculated automatically on the selected area (use of macros developed in-house) for each saved image.

The color change was homogeneous in diluted meat juice, and the variation in the redness coordinate in the CIELAB system, *a**, was correlated with the variation in ORP. Variations of ORP and *a** were also measured on muscle samples stored on air trays at low temperature, under different surface microbiological conditions: (1) without any preliminary treatment, (2) after a preliminary decontamination treatment, and (3) after decontamination followed by an inoculation treatment. The measurements were used to study the effect on *a** of four factors: two processing factors (storage temperature and oxygen content) and two non-processing factors (aging time and cutting direction relative to muscle-fiber orientation). Given that quadratic effects are known to be significant during meat oxidation, a Jones–Nachtscheim design was used to evaluate quadratic effects in addition to the linear effects. Each factor was studied with three levels, i.e., storage temperature at 2 °C, 6 °C, and 10 °C, oxygen content at 0%, 20%, and 100%, aging time at 0, 7 or 14 days, and cutting direction at 0°, 45° and 90°. Preliminary experiments showed that the beginning of meat browning due to myoglobin oxidation did not happen at the same time at every point on the ribeye steak surface. Hence, if changes to *a** were to be determined considering the entire meat surface, then the starting times of the color change would not be readily detectable, and the repeatability of the time-course curves for change in *a** would be negatively impacted. To solve this problem, different surface areas were tested to obtain repeatable measures and improve the detection of the starting time of meat browning. These areas were obtained by dividing the total number of pixels of half of the cut ribeye meat surface by 1, 2, 4, 16, 32, 64, and 128. From the 1/64 division onward, there was no further variation in the curve of *a**, and the starting time of the color change was clearly detectable. Therefore, *a** was measured on 1/64 of the total area at the location of color change. Note, however, that this specific area was located at the edge of the ribeye steak, near an aponeurosis that included fat tissue. During the experiments, the kinetics of *a** were found to vary in three phases: a lag phase that took place before browning and where *a** remained approximately equal to its initial value, a decreasing exponential phase, and then a final plateau where *a** reached its minimum value.

### 2.4. Development of the Mathematical Models

Two modeling approaches were compared to predict color variation in fresh beef meat during storage under different temperatures and oxygen contents in the packaging.

The first approach was phenomenological and assumed the shape of evolution by fitting the results to a predefined function.

The second approach based on the myoglobin oxidation model of Kondjoyan et al. [36] was more fundamental, while still making the following assumptions: the meat was modeled as a unidimensional single muscle compartment; the reaction rate constants, available at 20 °C, were adapted at the storage temperatures using a single activation energy value; the fraction of MbFe^4+^ was also neglected in comparison to the fractions of the other forms of myoglobin. Those assumptions established model limitations. Notably, this model does not simulate the food microstructure and, hence, does not consider the phenomena that exist at mitochondrial scale, such as the MMb reduction via the enzymatic NADH-dependent metmyoglobin reductase pathway or via the mitochondrial electron transport chain that may be important factors for oxidation.

Moreover, the reaction–diffusion model predicts chemical myoglobin states instead of color, while there is no simple relation in the literature between the concentrations of the myoglobin forms and the color coordinates measured in the CIELAB system. Therefore, we established relations on the basis of ratios between the different forms of myoglobin to predict color. Under this ratio approach, it was implicitly assumed that the absorption spectra of the three forms of myoglobin were sufficiently distant from each other. This assumption holds true for MMb and MbO_2_, as their absorption peaks are at 503 nm and 557 nm, respectively, even though they do share some overlap. It was also assumed that (i) variation of the chromatic dimensions of meat color in the CIELAB system depended essentially on *a**, and not on *b**, and (ii) at the highest MbO_2_/MMb or MbO_2_/(MMb + Mb) values, variations of these ratios led to a variation of lightness (*L**), with the variation in *a** happening later on. The first of these assumptions is consistent with the pattern observed when all three CIELAB coordinates were measured on fresh beef meat during storage. Indeed, the variation of *b** was irregular and much less substantial than the variation of *a**. The second assumption reflects the fact that the variation of *a** was only detectable after a clear lapse of time, while the variations of MbO_2_ or MMb started from the very beginning of time in storage. This assumption resulted in the existence of a limit value, *S*, for MbO_2_/MMb ratio or MbO_2_/(MMb + Mb) ratio, which separated the beginning of the variation of MbO_2_ and MMb from the beginning of the variation of *a**.

Note that a list of symbols is provided in Appendix A, Table A1, and that detailed presentations of the phenomenological model and of the reaction–diffusion model used in this paper are available in the literature [36,42].

#### 2.4.1. Phenomenological Model

The first approach was to apply the phenomenological model developed by Cucci [42] to directly predict the variation of *a** in ribeye steaks during storage. This model was based on data acquired in laboratory conditions following the previous experimental design. In a similar way to approaches employed in predictive microbiology, there are two steps in the modeling. First, a Gompertz model describes the shape of the change in parameter *a** as a function of two parameters, i.e., lag time and maximum rate of change, as well as initial and final values of *a** (Equation (1)).
(1)$$a^{*}=a^{*}{}_{0}+\left(a^{*}{}_{f}-a^{*}{}_{0}\right)e^{-e^{\frac{\mu_{max}*e^{1}*\left(lag-t\right)}{\left(\left(a^{*}{}_{f}-a^{*}{}_{0}\right)*\log\left(10\right)\right)+1}}}.$$

Then, a linear regression model provides the change in these two parameters as a function of the four experimental factors: oxygen content, storage temperature, aging time, and cutting direction (Equation (2) and (3)).
(2)$$lag=10^{b_{0}+b_{1}*pO_{2}{}^{2}+b_{2}*pO_{2}*Angle+b_{3}*T*pO_{2}+b_{4}*Angle^{2}+b_{5}*pO_{2}}.$$
(3)$$\mu_{max}=-10^{c_{0}+c_{1}*pO_{2}+c_{2}*pO_{2}{}^{2}+c_{3}*Mat^{2}+c_{4}*pO_{2}*Mat+c_{5}*Meat*Angle}.$$

Sensitivity analysis was performed on the parameters, and confidence intervals on the time to color shift were evaluated through Monte Carlo analysis, the detailed methodology and figures for which are detailed in Cucci [42]. All parameters were used in the analysis. However, *Angle* and *Mat* were found to have much less influence than the other factors [42]. Therefore, for the present study figures, cutting direction and meat aging were set at 0° and 14 days, respectively.

#### 2.4.2. Reaction–Diffusion Model

The second approach predicted the variation of *a** on the basis of the change in the three forms of myoglobin calculated by the reaction–diffusion model of Kondjoyan et al. [36]. This model solves a system of 22 oxidation reactions, with rate constants of different orders of magnitude, primarily influencing myoglobin chemical state (Equation (4)).
(4)$$\frac{d\left[C_{i}\right]}{dt}=\overset{N}{\underset{j=1}{\sum }}a_{ij}k_{j}\left(\underset{i\in \left[1,M\right],a_{ij}<0}{\prod }{\left[C_{i}\right]}^{-a_{ij}}\right).$$

For oxygen only, 1D diffusion from the surface to the inside of the meat is added to the reactions (Equation (5)).
(5)$$\frac{\partial \left[O_{2}\right]}{\partial t}=D_{O_{2}}\frac{\partial^{2}\left[O_{2}\right]}{\partial x^{2}}+\overset{N}{\underset{j=1}{\sum }}a_{O_{2}j}k_{j}\left(\underset{a_{O_{2}j}<0}{\prod }{\left[O_{2}\right]}^{-a_{O_{2}j}}\right).$$

In the previous study [36], the calculations of the myoglobin profiles were performed at a temperature of 20 °C. Here, myoglobin concentrations needed to be calculated at the surface at 2 °C, 6 °C, and 10 °C to compare the computations against the previous measurements of *a**. Thus, the system of equations was completed with an Arrhenius relation to determine the effect of temperature on the rate constants (Equation (6)). A single value of the activation energy *Ea* was considered for all the reactions, to decrease the number of parameters to be identified, and to avoid overfitting due to the relatively small amount of data in the experimental dataset.
(6)$$k_{jT}=k_{j20{}^{\circ}C}exp\left(\frac{-E_{a}}{RT}\right).$$

In the literature, variations in *a** measured on meat during storage are often only correlated with the increase in percentage of metmyoglobin. However, this is not enough, because the persistence of the bright red color of meat also depends on the concentration of MbO_2_ at the meat surface. Therefore, the variation of *a**/*a**_0_ (*a**_0_ being the initial value of *a**) was correlated to either the MbO_2_/MMb ratio or the MbO_2_/(MMb + Mb) ratio. As the sum of the three forms of myoglobin was assumed to be 100% (MbFe^4+^ was neglected, Equation (7)), the MbO_2_/(MMb + Mb) ratio was equal to MbO_2_/(1 − MbO_2_) and, thus, only depended on the proportion of MbO_2_ (Equations (8) and (9)).
MbO_2_ + MMb + Mb = 1.(7)
*a**/*a**_0_ = 1 for MbO_2_/(1 − MbO_2_) ≥ *S.*(8)
*a**/*a**_0_ = MbO_2_/(1 − MbO_2_) for MbO_2/_(1 − MbO_2_) < *S.*(9)

Preliminary calculations proved that the values of MbO_2_/(MMb + Mb) and MbO_2_/MMb were very close to one another. This can be explained by the fact that the percentage of Mb at the meat surface was very low right after its exposure to oxygen, i.e., just after the initial blooming period that immediately followed the opening of the vacuum-package. Below, the results are presented using only the MbO_2_/(MMb + Mb) ratio. The values for *Ea* and *S* were identified though least squares minimization between the measured and calculated *a**/*a**_0_. In our previous work [36], 200 numerical meshes were enough to calculate the profiles of the various forms of myoglobin in the meat cut. The computed results were consistent with the spatial distributions of the myoglobin forms measured [34] in LT meat cuts stored at 20° C. However, a 200-mesh grid was not enough to accurately determine the time-course change in color at the surface of the meat. Therefore, in this study, a more precise grid (400 meshes) was used to predict the time-course change in meat-surface color. As this denser grid increases calculation times, preliminary simulations were used to range the parameters of the model. During the identification process, calculations of the myoglobin forms were first performed on 100- to 200-mesh grids. To further accelerate the calculations, a substantially reduced oxidation reaction scheme that was shown to lead to the same values for MbO_2_ and MMb at the meat surface as the complete oxidation scheme [36] was used.

## 3. Results

Color variations were similar on untreated and decontaminated meat surface during the first 10–12 days of storage [42]. This was explained by the low level of contamination at the surface of untreated meat samples (about 10^2^ colonies CFU/cm^2^) at the beginning of storage in our experiments. The effectiveness of the probe to correlate ORP with variation of *a** in meat juice was validated by Cucci et al. [41]. This in-house probe was also used to compare variations of ORP with variation of *a** at the surface of meat samples. While the correlation between ORP and color remained, variations in ORP preceded variations in *a** contrary to what happened in liquids [42]. The presence of the oxidized iron rod promoted oxidation at the surface of meat; this is a limitation for its potential predictive use in the meat industry. The response time in meat was also slower than in liquid, further hampering industrial application.

The experimental results showed that, for an oxygen concentration of 20%, the decrease in product temperature increased the duration of the lag phase of *a** and decreased its exponential decay rate. These effects were more pronounced between 2 °C and 6 °C than between 6 °C and 10 °C. The increase in oxygen concentration from 20% to 100% had the same consequences when product temperature was 6 °C or 10 °C, whereas the effect of O_2_ was weaker when product temperature was 2 °C. Aging time and cutting direction had little effect compared to the primary factors: product temperature and O_2_ concentration. The following sections compare the variations of *a** measured on untreated beef meat samples against the variations of *a** during the first 15 days of storage predicted by mathematical models.

### 3.1. Identification of Parameters S and Ea in the Reaction–Diffusion Model

Least squares minimization between experimental and calculated results was performed based on the experimental kinetics of *a**/*a**_0_ measured by Cucci [42] for all available temperatures (2 °C, 6 °C, and 10 °C) and oxygen contents (20%, 100%). Ranges of *S* and of *Ea* were gradually narrowed using direct calculations and 100- or 200-mesh grids, leading to a range of optimal *S* value between 2 and 3. Then, the Levenberg–Marquardt algorithm was used to find the *Ea* values that minimized the residues (difference between model predictions and experimental values) considering a 100-mesh grid and two values for *S*, i.e., either 2 or 3. Finally, direct calculations were performed using a 400-mesh grid to calculate the residues with the previous ‘optimal’ *S* and *Ea* values. The least squares minimization procedure on the residues finally returned an optimum set of parameters: *S* = $2_{-0}^{+1}$ and *Ea* = 32.5 ± 2.5 kJ/mol. Table 1 presents the root of residual sum of squares obtained in preliminary calculations and for optimized parameters with the final 400-mesh grid.

Figure 2 shows the separate effect of variations of *S* and of *Ea* on the calculated ratio MbO_2_/(MMb + Mb) limited and normalized by its value at *S*, for a product temperature of 2 °C and an oxygen concentration of 20%. These normalized ratios were compared, using the 400-mesh grid, to the experimental kinetics of *a**/*a**_0_ measured by Cucci [42]. The calculated and the experimental normalized values evolved in a similar way. For a given value of *Ea*, the decrease in *S* value resulted in both a longer period where *a** remained constant and an increase in the rate of *Ea* decay during the following decay period. An increase in *Ea* value at constant *S* value also lengthened the stagnation period, albeit in a nonlinear way.

Analysis of the kinetics and the table of residues led us to use the parameters *S* = 2 and *Ea* = 32.5 kJ/mol and the 400-mesh grid for the final calculated results.

### 3.2. Effect of Temperature and Oxygen Content on the Kinetics of a* Calculated by the Models

Figure 3 illustrates the impact of temperature and package atmosphere oxygen content on the kinetics of *a** calculated by the reaction–diffusion model for the extreme temperature and oxygen conditions of the experimental design. The increase in temperature significantly reduced the predicted color shelf-life, consistently with the measurements and with common sense. The limit value of *a**/*a**_0_ at which consumers start to reject the product remains a subject of debate and may show individual or cultural consumer variability; hence, the use of laboratory measurements to exactly predict the consumer acceptance of meat is known to be difficult [43]. Nevertheless, the trends are the same whatever the *a**/*a**_0_ limit value chosen. The 100% oxygen condition increased the model-predicted time of meat color change, thus delaying consumer rejection, by one to a few days at 2 °C. These few days of longer shelf-life until rejection remained when temperature was 10 °C. Moreover, the increase in shelf-life due to O_2_ variation from 20% to 100% was greater at 10 °C than at 2 °C.

Figure 4 compares the results calculated by the two models when the meat was stored at 2 °C and 6 °C under 20% oxygen. This is an important case scenario, because it corresponds to the set of conditions most commonly encountered in practice. The models led to similar results even if there were some notable differences. The initial decrease in *a** from *a**_0_ was gradual when using the phenomenological model but abrupt when using the reaction–diffusion model. This is directly related to the form of the Gompertz equation used in the phenomenological model, and to the assumption of a single threshold value in the case of the reaction–diffusion model. Experimental uncertainties make it impossible to know which of these two evolutions is the closest to reality. However, it is likely that the decrease in *a**/*a**_0_ from 1.0 is not as abrupt as observed with the reaction–diffusion model, and that there is not a single threshold value but a window of threshold values to be used in the reaction–diffusion model. The decrease in the curves for values of *a**/*a**_0_ lower than 20% was more marked when using the phenomenological model than the reaction–diffusion model. This can be explained both by greater measurement uncertainties when *a** approaches 0 and by the use of a single *Ea* value for all the oxidation reactions in the reaction–diffusion model. Despite these differences, it could be concluded that both the phenomenological model and the reaction–diffusion model predict similar values when *a**/*a**_0_ varied between 80% and 40%. It turns out that it is this portion of the curves that has consequences in terms of color stability and, thus, shelf-life, as *a**/*a**_0_ values that are either higher than 80% or lower than 40% systematically lead to acceptance or rejection of the product by consumers.

## 4. Discussion and Prospects

This section discusses the prospects for improving the models and sensors and for generalizing the approaches presented in the paper to meats other than beef and to fish-based foods.

### 4.1. Prospects for the Modeling Approaches

Phenomenological models have advantages and limitations. They are easy to build, their calculation times are short, and they can prove powerful solutions for controlling a process or determining optimal storage conditions if both product and process are standardized. However, the phenomenological model used here only describes what happens at the surface of the meat; hence, it is unable to predict oxidations that take place under this surface. Indeed, when in-package oxygen content is very high, the meat surface may still appear bright red while the meat below the surface is brown [36,44]. A phenomenological model is also unable to predict situations that fall completely outside the experimental design used to build it. It, therefore, cannot anticipate variations in product composition that were not within the experimental domain. Thus, for beef, it would theoretically be necessary to repeat the experiments on the different muscles of the carcass while factoring for breed, sex, age, and diet of the animals, which would require a prohibitively high number of experiments. To go further in the prediction of meat browning, it is, therefore, important to use more fundamental models, such as reaction–diffusion models. This paper demonstrated how a reaction–diffusion model can be used to predict meat color. The performance of this reaction–diffusion model can still be improved. The calculation time can be shortened by using progressive mesh refinement near the surface of the meat, as well as other advanced numerical techniques. Its accuracy could be improved by varying the *Ea* parameter from one reaction to another. This reaction–diffusion model also has mechanistic limitations. Notably, it is unable to predict the oxidation mechanisms at all scales of the food structure, and the one-dimensional, single-compartment nature of the model does not consider the multiscale fat compartments. However, the model can be improved to overcome these drawbacks. At the scale of the meat cut, combining two-dimensional oxygen transfer and the greater solubility of oxygen in fat than in meat [45] would make it possible to numerically simulate the oxidation that often occurs near the superficial fat cover or near the fat separations between different muscles of a ribeye steak. At a smaller scale, the effect of small fat particles in the lean meat compartment (beef marbling) and lipid oxidation inside these particles could be modeled using a single equivalent ‘fat compartment’ that would exchange reactants with the lean meat compartment. Similarly, a single equivalent mitochondrial compartment could be added in the model to account for the oxidation mechanisms that occur at this scale.

In this paper, the effect of microorganisms on surface oxidation was neglected, as the initial level of microbial contamination was about 10^2^ CFU/cm^2^ at the beginning of storage. However, the surface of carcasses stored in slaughterhouses can count up to 10^4^ CFU/cm^2^, in which case the influence of bacteria on surface oxidation should be considered. In the phenomenological model, the equations chosen to describe the effect of chemical oxidation on color had the same mathematical form as the equations used in predictive microbiology; thus, it would theoretically be easy to add other equations to predict the effect of microbial growth rates. However, this would also require a much more complete experimental design to separate the effects related to chemical oxidation from those related to microbiology. It would be instructive to compare this approach with results obtained through the reaction–diffusion model that clearly separates the effects of each of the chemical and microbiological reactions. To go further, bridges could be created between the phenomenological model and the reaction–diffusion model. The present mathematical form of the phenomenological model can be changed for a simplified Turing reaction–diffusion system that can consider diffusion, oxidation reactions, and microbial development at the same time and in different compartments of the food. Both types of models—the more complete reaction–diffusion model and the phenomenological model—could then be used in synergy with optimized experimental designs. This would allow both testing fundamental hypotheses on the role of product composition, microbial flora, and transfer mechanisms on oxidation in food products and developing more robust control models that are transferable to industry.

### 4.2. Prospects for Sensors

Attempts to develop new sensors prove that it remains difficult, even under laboratory conditions, to measure ORP at the surface of solid foods without influencing the product. The developed sensor confirmed the correlation between ORP and color for both meat and meat juice; however, its practical application is limited by its slow response time in meat and the oxidation-promoting impact of the oxidized iron rod. Oxidized iron was chosen as it is the natural evolution of iron in meat conditions and there is little further evolution to expect. In order to prevent the oxidation-promoting impact, other reductants may be considered to replace this electrode, e.g., aluminum or zinc. However, this would favor the creation of other ionic compounds susceptible to reacting with the meat in other chemical systems, different from the iron chemistry. There is, therefore, little prospect for applying this type of sensor under industrial conditions. However, tests carried out in liquids can lead to a better understanding of the effect of specific bacterial populations on change in ORP in relation to change in color. Results on this aspect have already been obtained [42] but are not detailed here. This type of work would be instructive for biopreservation, which uses microorganisms seeded in the food to limit the growth of pathogens without degrading the product. For example, probes could be used to study the impact of different bacterial strains used for biopreservation on the oxidation of the product. The results would increase our knowledge on the effect of bacteria on oxidation and could then be introduced into the models to improve their predictive capacity. Further improvement would come from advances in analysis of the kinetics of the oxidation reactions. It could also be useful to test the response of new sensors in minced and pasty food matrices, which are used to develop new formulated or textured foods that are potentially exposed to significant oxidation phenomena. In this case, it would be important to cross-check the response of the ORP sensors with that of electronic noses, especially when off-flavors are associated with the defect of color in the rejection of the food by consumers [46].

### 4.3. Extension to Other Meats and Fish

This paper applied modeling to the variation of beef meat color due to myoglobin oxidation. However, as oxidation also drives color degradation in food products other than beef, it would be useful to analyze whether the models and sensors developed in this paper can be generalized to other raw meat and fish foods. Beyond beef, the reaction–diffusion model could be applied to lamb [10,25,32], pork [4,15,33], or fish [8,21,46], such as bluefin tuna [9,27]. In this case, the reaction–diffusion model developed on beef would have to be completed to consider the specific compositional profile of tuna meat and especially the oxidation of its specific unsaturated acids. Other specific degradation reactions connected to the specific biochemistry and spoilage microflora of fish have to be considered, such as the reduction of trimethylamine oxide in its nitrogenous derivatives, which was studied in the situation of the development of off-flavors during the storage of fresh sea bass [46]. Differences in freezing temperatures after the fishing of wild tuna can also be predicted if the *Ea* values of the most color-impacting reactions, particularly the lipid oxidation reactions, are determined at these negative temperatures. The reaction–diffusion model developed in this paper could also be generalized to the preservation of ground meat products, if the incorporation of oxygen and its specific diffusivity values in ground matrices are added in the model, and if the effect on oxidation of the microorganisms mixed with the ground meat is considered. When ground meat is stored under MAP, the diffusion of CO_2_ and its effect on microbial growth would also need to be included in the reaction–diffusion model. When oxidation is counteracted by adding antioxidants to the mix [13,14,15,16,17,18,19,20,21,22,23], the antioxidant reactions have to be added to the scheme, while, if antioxidant seeds are directly deposed on the surface of meat pieces, such as in the case of minced fish [24], simplified compartmental reaction–diffusion models could be used to consider the effect of the antioxidant film on oxidation of the minced fish.

## 5. Conclusions

This paper aimed to discuss how new modeling approaches and sensors can be used to prevent the loss and waste of raw meat and fish foods due to oxidation during storage. The case study was the prevention of biochemical oxidation-induced browning of beef meat cuts. Storage temperature and oxygen content in the environment are the main factors affecting this undesirable browning. A phenomenological model was built to describe the time-course of the redness coordinate *a** (CIELAB system) at a given location of the surface of beef ribeye steaks in different temperature and oxygen content conditions. The model was able to predict the experimental evolution of *a**. However, it remains limited to the variation of *a** at a given location at the surface of the ribeye steak and to the range of temperature and oxygen considered in the experimental design. To go further, a reaction–diffusion model was used to predict color change from the myoglobin oxidation reactions. It was possible to correlate the variation of *a** to the MbO_2_/MMb ratio or MbO_2_/(MMb + Mb) ratio calculated from the results of the reaction–diffusion model using a few assumptions. Furthermore, a new sensor was developed to measure oxidation-reduction potential (ORP) in meat juice and at the surface of the *LT* muscle. The evolution of ORP measured by the sensor, and the measurements of *a** were tightly correlated in homogeneous liquid conditions but not on meat surface, where ORP measurement remains a bottleneck. Oxidation causes degradation of quality in many foods other than beef meat cuts. The models developed in this paper can be applied to pieces of lamb, pork, or some fish rich in myoglobin, such as bluefin tuna. However, those extensions would most often depend on the addition to the reaction scheme considered in this paper of lipid oxidation reactions and of the effect of surface microorganisms on oxidation. The sensor and the modeling approaches developed in this paper can also be applied to ground and restructured meat and fish foods that are potentially subject to extensive oxidation phenomena. In this case, the model can be adapted by adding the effect on oxidation of the microorganisms mixed within the product, the specificity of gas diffusion in ground matrices, and, in some cases, geometrical compartmentation. This can pave the way to the design of innovative and more sustainable foods where ORP sensors and models can provide complementary inputs.

## Figures and Tables

**Figure 1 foods-11-01139-f001:**
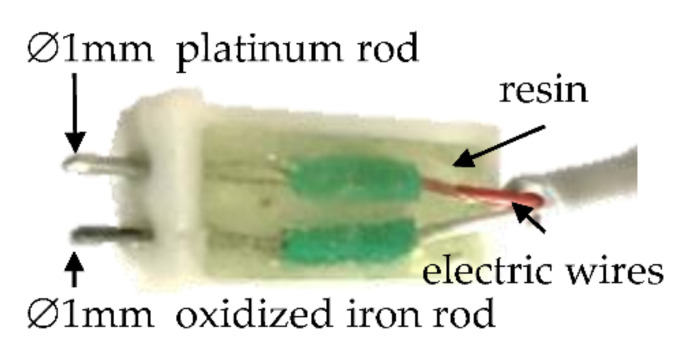
Design of the oxidation–reduction potential (ORP) sensor.

**Figure 2 foods-11-01139-f002:**
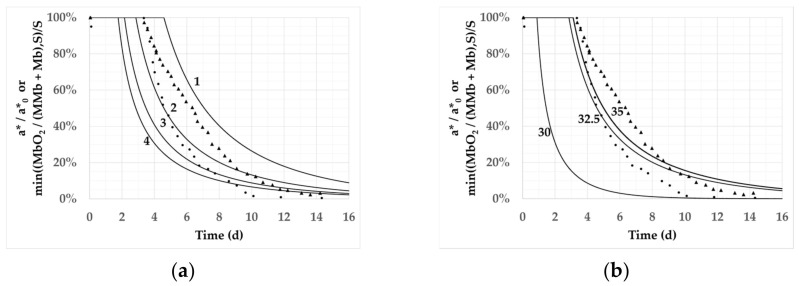
Sensitivity of the reaction–diffusion model to its main parameters. Experimental data for *a**/*a**_0_ at T = 2 °C and 20% O_2_ from Cucci [42] displayed with circles (first repetition) and triangles (second repetition). Solid lines plot the value of the ratio MbO_2_/(MMb + Mb), normalized by its value at *S*, and calculated from the reaction–diffusion model with T = 2 °C and 20% oxygen. (**a**) Impact of the threshold value *S* (equal to 1, 2, 3, or 4) on the predictions when *Ea* = 32.5 kJ/mol; (**b**) impact of *Ea* on predicted outcome (*Ea* = 30, 32.5, or 35 kJ/mol) when S = 2.

**Figure 3 foods-11-01139-f003:**
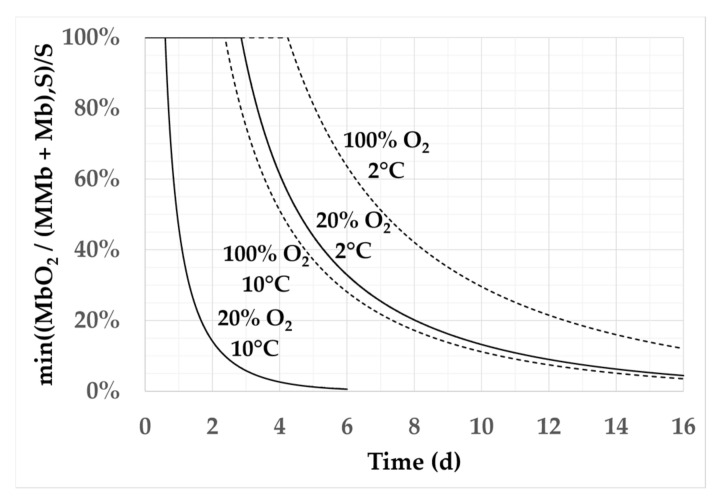
Impact of environment conditions on the model outputs. Impact of the packaging oxygen content and storage temperature on the normalized MbO_2_/(MMb + Mb) results calculated by the reaction–diffusion model with *S* = 2 and *Ea* = 32.5 kJ/mol. Solid lines and dashed lines plot 20% and 100% O_2_, respectively.

**Figure 4 foods-11-01139-f004:**
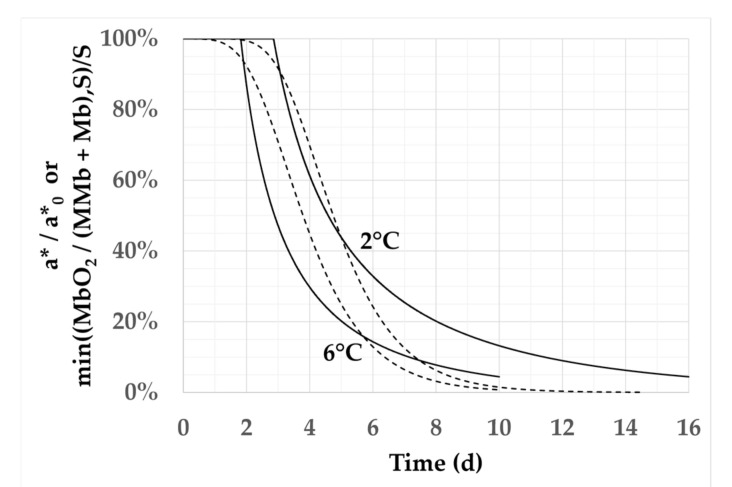
Impact of temperature on the model outputs. Comparison of the results of the phenomenological model (dashed lines) and the reaction–diffusion model with *S* = 2 and *Ea* = 32.5 kJ/mol (solid lines) at 20% O_2_ and *T* = 2 °C and 6 °C.

**Table 1 foods-11-01139-t001:** Reaction–diffusion model quality of fit ^1^ for all environmental conditions.

**S**		2	2	3	3	2
***Ea* (kJ/mol)**		30	35	30	35	32.5
**Mesh**		100	100	100	100	400
**O_2_**	**T (°C)**	**Roots of the residual sum of squares**				
20%	2	0.37	0.34	0.39	0.42	0.09
	6	0.27	0.24	0.30	0.33	0.20
	10	0.39	0.36	0.43	0.46	0.27
100%	2	0.17	0.26	0.15	0.13	0.20
	6	0.13	0.11	0.17	0.22	0.11
	10	0.16	0.13	0.21	0.25	0.15
**Mean**		0.25	0.24	0.28	0.30	0.17

^1^ Root of the sum of the squares of the residues between the experimental observations for all temperatures and oxygen contents in the experimental design of Cucci [42] and the results calculated by the reaction–diffusion model around the optimal values of parameters *S* and *Ea*. The residues were calculated first with the coarse 100- or 200-mesh grids and finally with the 400-mesh grid.

## Data Availability

Data is contained within the article.

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
