# Peer review of "Predicting the Oxidative Degradation of Raw Beef Meat during Cold Storage Using Numerical Simulations and Sensors—Prospects for Meat and Fish Foods"

_foods, 2022, doi:10.3390/foods11081139_

Round 1

Reviewer 1 Report

Comments to the Author

MANUSCRIPT DETAILS

Ms. Ref. No.: foods-1643672-peer-review-v1

Title: Preventing the oxidative degradation of raw beef meat during cold storage using numerical simulations and sensors – Generalization to meat and fish foods

Article Type: Article

JOURNAL: Foods

GENERAL COMMENTS

This manuscript aimed develop a new sensor and modeling approaches to predict the oxidative browning of beef rib steaks stored under different temperature and oxygen concentration conditions.

This MS main idea is interesting as it is based on developed “numerical simulations and sensors” detecting and monitoring meat oxidation. So, its main pillar should be clear and precise Materials and Methods description. Authors should re-write/ design Methods section illustration to serve this point.

I encourage authors to consider the comments below to put their research work in better form.

Title

The MS aims to “Predicting” rather than “Preventing”, I suggest modification to avoid misleading title.

Abstract

- L15: Food waste, revise thoroughly.

- L17: “Measurement of redox potential (ORP)”, correct either to Redox potential (Eh) or to Oxidation-reduction potential (ORP). Revise thoroughly.

- Provide some significant values from your results in Abstract section

Introduction

Introduction section is insufficient. Kindly, rewrite to cover a decent background of you research work supported with recent references.

Materials and Methods

- This section is written in a confusing way, it needs to be re-written precisely, clearly and properly to provide;

  • Sufficient description of meat samples (dimensions, sources, initial/ storage conditions..)
  • Collected approach assumptions (Table 1, L187) and then detailed description
  • Full description of the “in-house probe” designed by authors and a photo/diagram that clearly illustrate/ compare the devolvement(s) conducted
  • Support data with sensory analyses may help to connect provided data with organoleptic properties
  • Simple oxidation chemical analyses e.g. protein or lipid oxidation can help indicating the variations in color due to oxidation    

- Methods section should describe the approaches and procedures conducted by authors to achieve their aim. So, L52-65 and similar data preferred be provided in Introduction section.

- “We” e.g. L56, 62, 65.. is not preferred to be used. Kindly, revise thoroughly and modify accordingly.

- Revise Methods section to provide necessary references that guided authors in developing their models.

- L83: “The following sections compare..”, kindly, rearrange to put each section data all together without repeating

Results

- Chose suitable precise Tables’/ figures’ titles, then describe details in footnotes or figures’ legends.

- Provide better resolution of figures

Discussion

- Discussion section needs to be rewritten based on redesigned Methods

- L377: “Extension to other meats and fish”, did authors used fish samples? This section is for discussing obtained data for already conducted experiments, then based on these data authors can recommend other applications in Conclusion as “recommendations”

References

- References need update to 2022.

Author Response

General comment on our paper.

This paper is dedicated to a Foods special issue on “Emerging Technologies in the Food Industry: Sustainability Assessment and Consumer Acceptability”. It aims at illustrating how mathematical models and new sensors can predict the oxidative degradation of meat and fish products, and thus help to prevent animal source waste. Even if the modelling and experimental comparisons have only been done on beef meat, it is to our point of view a starting point to discuss the generalization of the approach to other meats than beef meat and to some fish products. This objective to go further in the analysis that the only case of beef meat had led to an initial version of the manuscript that was not clear and had to be revised.  As asked by the reviewers, in the new version of the manuscript the references to literature, and the definitions, have been grouped in the “Introduction”, the “material and methods” and “results” sections have been detailed and present in a more conventional way, and the conclusion have been shortened. We have maintained a shorter and more structured “Discussion and prospects” section to analyze how the approach followed on beef meat can be applied to other meats than beef meat, and to some fish products, while trying to be clear that no modeling or experimental works have been done in this case.

REVIEWER#1

Title

The MS aims to “Predicting” rather than “Preventing”, I suggest modification to avoid misleading title. Modified.

Abstract

- L15: Food waste, revise thoroughly. Corrected in the whole manuscript.

- L17: “Measurement of redox potential (ORP)”, correct either to Redox potential (Eh) or to Oxidation-reduction potential (ORP). Revise thoroughly. Changed for oxidation-reduction potential (ORP) in the whole manuscript.

- Provide some significant values from your results in Abstract section

Added “Both models predicted similar storage durations for acceptable colour, although deviating for higher and lower redness levels, which are of no interest for meat acceptance. Simulations under higher oxygen concentrations lead to a few days of delay in the redness change, as observed in practice, under modified atmosphere packaging. In meat juice, variation in ORP measured by the sensor correlated with the redness variation. However, in meat, sensors promote the oxidation at the adjacent area, which is unacceptable for industrial use.” results to the abstract section L20-25.

Introduction

Introduction section is insufficient. Kindly, rewrite to cover a decent background of you research work supported with recent references. Introduction has been rewritten and supported by recent references.

Materials and Methods

- This section is written in a confusing way, it needs to be re-written precisely, clearly and properly to provide;

  • Sufficient description of meat samples (dimensions, sources, initial/ storage conditions..) Added section 2.1 L107-122 on meat samples.
  • Collected approach assumptions (Table 1, L187) and then detailed description. Approach assumptions have been grouped at the beginning of section 2.4 L181-215 on development of the mathematical models.
  • Full description of the “in-house probe” designed by authors and a photo/diagram that clearly illustrate/ compare the devolvement(s) conducted. A description of the probe have been added in section 2.2 L124-132. Reference to already published visual was added.
  • Support data with sensory analyses may help to connect provided data with organoleptic properties. The relation here is between colour and chemical oxidation; colour being characterized by instrumental measurements, less dependent on individual experimenter variation. Added section 3.2. L340-343 : The limit value of a* / a*0 at which consumers start to reject the product remains a subject of debate, and may show individual or cultural consumer variability, and the use of laboratory measurements to predict exactly the consumer acceptance of meat is known to be difficult [40].
  • Simple oxidation chemical analyses e.g. protein or lipid oxidation can help indicating the variations in color due to oxidation. Prediction of beef myoglobin oxidation by the reaction-diffusion model was compared in a previous work to measurements [36]. This model is assumed here to be valid and the paper focuses on the transition from myoglobin predictions to colour predictions, which are compared to colour measurements.

- Methods section should describe the approaches and procedures conducted by authors to achieve their aim. So, L52-65 and similar data preferred be provided in Introduction section. Done

- “We” e.g. L56, 62, 65.. is not preferred to be used. Kindly, revise thoroughly and modify accordingly. Text has been modified to remove the “We”.

- Revise Methods section to provide necessary references that guided authors in developing their models. The Methods section has been thoroughly revised; with clearer referencing of models origins and validation.

- L83: “The following sections compare..”, kindly, rearrange to put each section data all together without repeating Done

Results

- Chose suitable precise Tables’/ figures’ titles, then describe details in footnotes or figures’ legends. Added concise titles to figures, complementary to the detailed description.

- Provide better resolution of figures Done

Discussion

- Discussion section needs to be rewritten based on redesigned Methods Done

- L377: “Extension to other meats and fish”, did authors used fish samples? This section is for discussing obtained data for already conducted experiments, then based on these data authors can recommend other applications in Conclusion as “recommendations”. No modeling or experimental works have been done on other meats than beef meat, and on fish products.

However, we have maintained a shorter and more structured “Discussion and prospects” section to analyze how the approach followed on beef meat can be applied to other meats than beef meat, and to some fish products. The main outcomes of this analysis have been summarized at the end of the “Conclusion”. See the “General comment on our paper” written above.

  References

- References need update to 2022. Recent references were added through the manuscript.

Reviewer 2 Report

It is necessary to attend to the comments (document) to improve the structure of the work and make it fully understandable.

The manuscript “Preventing the oxidative degradation of raw beef meat during cold storage using numerical simulations and sensors – Generalization to meat and fish foods” is an interesting work, however, the writing throughout the text is not adequate. The writing looks like a story, with too much unnecessary or out of context information that needs to be synthesized and reorganized. It is necessary to attend to the comments (document) to improve the structure of the work and make it fully understandable.

Title:

The title is adequate; however, the direct application was to beef meat and therefore the “generalization to meat and fish foods” part should only be addressed in the perspectives or conclusions.

Abstract:

The abstract is adequate; however, it is necessary to point out whether the sensor associated with the proposed models worked or not.

Introduction:

The introduction has the necessary information to understand the topic, however, part of the fundamentals/definitions explained in the materials and methods section can be included in this section.

Materials and methods:

The methods are over developed since fundamentals and definitions are included and not just the method. THE FUNDAMENTALS, DEFINITIONS, JUSTIFICATIONS OR RESULTS ABOUT THE EXPERIMENTS SHOULD NOT BE EXPLAINED. A restructuring is necessary, for example:

2.1. Materials (it is necessary to mention the types of meat and their origin and the materials of the sensor (if relevant))

2.2. Study model (it is necessary to explain how many samples were analyzed and in which areas the measurements/or segments for the studies were taken)

  1. Manufacture of sensors (it is necessary to explain how I made the sensor)

2.4. Color measurements (it is necessary to explain how the measurements were made with the manufactured sensor and the parameters involved such as oxygen and temperature)

2.5. Development of mathematical models (it is necessary to explain the equations/model in general terms, for example, if the model determines the correlation coefficient, it must be explained that this correlation is based on the sample or population means).

Table 1. It must be developed and described in the text, as well as the proposed equations.

Table 2. It must be developed and described in the text, as well as the proposed equations.

Lines 113-201. They can be included (concisely) in the proposed section 2.4.

2.6. Design of experiments used.

Results:

There is a lack of organization in the results because they are confused, it should be improved.

Figure 1. Needs improvement due to poor quality, small print, and unreadable.

Figure 2. Needs improvement due to poor quality, small print, and illegible.

Figure 3. Needs improvement due to poor quality, small print, and unreadable.

Analysis:

This section is written in a colloquial way, and not much is said about the analysis of the results, but about perspectives and approaches towards other dietary models.

Conclusions:

This section is too long, it would be understandable if the information from lines 423 to 442 were kept, the rest is information that can be included in the perspectives and analysis.

Author Response

General comment on our paper.

This paper is dedicated to a Foods special issue on “Emerging Technologies in the Food Industry: Sustainability Assessment and Consumer Acceptability”. It aims at illustrating how mathematical models and new sensors can predict the oxidative degradation of meat and fish products, and thus help to prevent animal source waste. Even if the modelling and experimental comparisons have only been done on beef meat, it is to our point of view a starting point to discuss the generalization of the approach to other meats than beef meat and to some fish products. This objective to go further in the analysis that the only case of beef meat had led to an initial version of the manuscript that was not clear and had to be revised.  As asked by the reviewers, in the new version of the manuscript the references to literature, and the definitions, have been grouped in the “Introduction”, the “material and methods” and “results” sections have been detailed and present in a more conventional way, and the conclusion have been shortened. We have maintained a shorter and more structured “Discussion and prospects” section to analyze how the approach followed on beef meat can be applied to other meats than beef meat, and to some fish products, while trying to be clear that no modeling or experimental works have been done in this case.

REVIEWER #2

The manuscript “Preventing the oxidative degradation of raw beef meat during cold storage using numerical simulations and sensors – Generalization to meat and fish foods” is an interesting work, however, the writing throughout the text is not adequate. The writing looks like a story, with too much unnecessary or out of context information that needs to be synthesized and reorganized. It is necessary to attend to the comments (document) to improve the structure of the work and make it fully understandable. See our general comment, the manuscript has been synthesized and reorganized according to recommendation.

Title:

The title is adequate; however, the direct application was to beef meat and therefore the “generalization to meat and fish foods” part should only be addressed in the perspectives or conclusions. See our general comment on the reorganization of the paper

Abstract:

The abstract is adequate; however, it is necessary to point out whether the sensor associated with the proposed models worked or not. The sensor worked for its purpose of correlating ORP with color. However, it had limits hampering industrial use, which we initially presented inappropriately.

Added to abstract: In meat juice, variation in ORP measured by the sensor correlated with the redness variation. However, in meat, sensors promote the oxidation at the adjacent area, which is unacceptable for industrial use.

Furthermore, section 2.2 L123-132 now describes the sensor and section 3 L281-288 explains its effectiveness yet limits and provides the corresponding reference.

Introduction:

The introduction has the necessary information to understand the topic, however, part of the fundamentals/definitions explained in the materials and methods section can be included in this section. Introduction has been rewritten and all the definitions and the references to literature have been grouped in this part.

Materials and methods:

The methods are over developed since fundamentals and definitions are included and not just the method. THE FUNDAMENTALS, DEFINITIONS, JUSTIFICATIONS OR RESULTS ABOUT THE EXPERIMENTS SHOULD NOT BE EXPLAINED. Definition and justification have been moved to introduction.

A restructuring is necessary, for example:

2.1. Materials (it is necessary to mention the types of meat and their origin and the materials of the sensor (if relevant)) Done, L106-122.

2.2. Study model (it is necessary to explain how many samples were analyzed and in which areas the measurements/or segments for the studies were taken) Done in 2.3 L162-177.

  1. Manufacture of sensors (it is necessary to explain how I made the sensor) Done in 2.2 L123-132.

2.4. Color measurements (it is necessary to explain how the measurements were made with the manufactured sensor and the parameters involved such as oxygen and temperature) Done in 2.3 L134-161.

2.5. Development of mathematical models (it is necessary to explain the equations/model in general terms, for example, if the model determines the correlation coefficient, it must be explained that this correlation is based on the sample or population means). Mathematical models methods section have been restructured to group hypothesis at the beginning of the section, and the validation procedure have been (briefly, as a complement to referenced work) explained.

Table 1. It must be developed and described in the text, as well as the proposed equations. Done

Table 2. It must be developed and described in the text, as well as the proposed equations. Done

Lines 113-201. They can be included (concisely) in the proposed section 2.4. This portion have been redistributed into the new structuration for Methods.

2.6. Design of experiments used. Added in section 2.3 L157-161

Results:

There is a lack of organization in the results because they are confused, it should be improved.

Figure 1. Needs improvement due to poor quality, small print, and unreadable. Figure 2. Needs improvement due to poor quality, small print, and illegible.

Figure 3. Needs improvement due to poor quality, small print, and unreadable.

Size of print and quality were improved for figures 1-3.

Analysis:

This section is written in a colloquial way, and not much is said about the analysis of the results, but about perspectives and approaches towards other dietary models. Section rewritten

Conclusions:

This section is too long, it would be understandable if the information from lines 423 to 442 were kept, the rest is information that can be included in the perspectives and analysis. Conclusion has been shortened following the recommendation of the reviewer.

Round 2

Reviewer 1 Report

MANUSCRIPT DETAILS

Ms. Ref. No.: foods-1643672

Title: Preventing the oxidative degradation of raw beef meat during cold storage using numerical simulations and sensors – Generalization to meat and fish foods

Article Type: Article

JOURNAL: Foods

I appreciate the authors’ efforts responding to most of the comments, but the MS still needs some minor revisions.

Kindly, check the following comments:

- The MS needs language polishing especially in grammar, kindly revise thoroughly.

- Kindly provide a guide reference for sample preparation.

- Authors stated that they designed “in-house probe”, and stated in revised version that “the values and variations of the normalized ORP of our in-house probe were the same as for commercial ORP probes (Hanna Instruments, RI)”.

As previously requested; kindly provide a photo/diagram for your designed probe that reflects the innovative idea that differs from the commercial probe.

- Table (1), Precise the Table’s title and then move the info to Table’s footnotes.

Wish you all the best…          

Author Response

- The MS needs language polishing especially in grammar, kindly revise thoroughly.

We have revised thoroughly the manuscript.

- Kindly provide a guide reference for sample preparation.

In the section 2, we have completed the paper with regard to the identification of the Longissimus muscle and to the English name of the steak used during the experiments.  We have added some references on the identification of beef muscles and a guide about meat preparation.

- Authors stated that they designed “in-house probe”, and stated in revised version that “the values and variations of the normalized ORP of our in-house probe were the same as for commercial ORP probes (Hanna Instruments, RI)”.

As previously requested; kindly provide a photo/diagram for your designed probe that reflects the innovative idea that differs from the commercial probe.

We have added a photo of the new probe in a new figure1. We have added the following paragraph to section 2.2 to explain how the probe we had designed differs from the commercial probe:

“Commercial ORP sensors commonly use gels or electrolyte solutions, a platinum sensing pin, a ceramic junction (salt bridge), and a polyetherimide resin or glass body. Those designs make them suitable for measurement in liquids, but penetration into meat gives unreliable values and a risk of breakage. On the probe designed by us, both electrodes can safely penetrate into meat.”

- Table (1), Precise the Table’s title and then move the info to Table’s footnotes.

Done

Wish you all the best…   

Thank you for your helpful suggestions.

Reviewer 2 Report

The manuscript “Preventing (Predicting) the oxidative degradation of raw beef meat during cold storage using numerical simulations and sensors – Generalization to meat and fish foods” has been greatly improved, but some important changes need to be made.

Author Response

Title:
The final part of the title "generalization to piss and fish foods" converts to very general work, the sensors were only tested on beef and in the first part of the title it was very well established.

We want to maintain this idea in the title as an important point of the paper.  Indeed, even if the modelling and experimental comparisons have only been done on beef meat, it is to our point of view a starting point to discuss how our approach can be applied to other meats than beef meat and to some fish foods. We believe it to be coherent with the publication of this paper in a Foods special issue on “Emerging Technologies in the Food Industry: Sustainability…” where it seems possible to go further than only describing a experimental or modelling work to discuss some emerging prospects. However, we agree that “generalization” may be too strong in regards to the manuscript content, hence, in the title we have replaced “Generalization to meat and fish foods” by “prospects for meat and fish foods”.

Abstract:
The abstract is adequate; however, it is necessary to point out that the project was developed experimentally in beef.
Line 27. “and fish” should be removed

Done.

Introduction:
Lines 292-538. The paragraph looks like a conclusion, the order can be reversed to give it a sense of introduction, for example, “The development of artificial intelligence a form of modeling, has driven and continues to drive substantial advances in many fields of science and technology. However, efforts to predict oxidation in foods do not stand to benefit much from this development due to the small amount of data available on oxidation kinetics in storage conditions. Therefore, the paper will focus here on two other modeling approaches: phenomenological models and reaction–diffusion models. This paper shows how sensors that measured the ORP and mathematical models can help to better predict and thus prevent the oxidation of beef meat cutsresponsible for undesirable color change. The paper also discusses how this approach and the results obtained could be generalized to reduce waste and loss other raw meats and raw fish”.

Changed according to the suggestions.

Line 533. The experimentation allows some assertions to be made based on the phenomenon analyzed, but to argue that the work is applicable to other types of food models, a further argument is necessary. It is necessary to change the phrase "could be generalized to reduce waste and loss of other raw meats and raw fish" by "can be applied to other meats and raw fish".

Changed according to the suggestion.

Materials and methods:

The description of the methods improved, however, there is still overdeveloped information.
Line 542-544. They can be synthesized (the rest of the information is not necessary): “Rib steaks
from young cattle (Puigrenier slaughterhouse, Montluçon, France) were semi-dressed, vacuumpacked, and stored in the refrigerator at 4 ± 2°C”.

Changed according to the suggestion.

Lines 545-557. These lines can be synthesized.

We have modified the Section 2.1 to avoid repetitions and unneeded information.

Lines 559-567. They can be synthesized: “An in-house oxidation-reduction potential was designed and built with two 1-mm diameter 'working' electrodes, one is a platinum rod (99.95% purity, Surepure Chemetals, NJ) and the other an oxidized iron rod [38]. The rods were connected to a digital data acquisition device using electric wires with a gap between the electrodes, that was molded in resin. The electric signal was checked in various liquids and results were compared with the commercial probes (Hanna Instruments, RI)”.

Changed according to the suggestion. Note that we have also provide complementary information to answer to some comments of reviewer 1.

Lines 568-911. These lines can be synthesized.

We have modified these lines to avoid repetitions and unneeded information.

Results:
The results are adequate.

Discussion and prospects:
The prospects are adequate, the discussion could improve.

We have modified Section 4 as much as possible to shorten it a little bit and to improve its clarity.

Conclusions:
The conclusions are correct.

Thank you for your helpful suggestions.